# Antibiotic Resistance and Genetic Profiles of *Vibrio parahaemolyticus* Isolated from Farmed Pacific White Shrimp (*Litopenaeus vannamei*) in Ningde Regions

**DOI:** 10.3390/microorganisms12010152

**Published:** 2024-01-12

**Authors:** Fangfang Zhang, Jie Zhang, Guowen Lin, Xiaoqiang Chen, Huizhen Huang, Chunxia Xu, Hai Chi

**Affiliations:** 1Mindong Fishery Research Institute of Fujian Province, Ningde 352100, China; 2East China Sea Fisheries Research Institute, Chinese Academy of Fishery Sciences, Shanghai 200090, China

**Keywords:** *Litopenaeus vannamei*, *Vibrio parahemolyticus*, antibiotics resistance, virulence genes, resistance genes

## Abstract

To better understand the antibiotic resistance, virulence genes, and some related drug-resistance genes of *Vibrio parahaemolyticus* in farmed pacific white shrimp (*Litopenaeus vanname*i) in Ningde regions, Fujian province, we collected and isolated a total of 102 strains of *V. parahaemolyticus* from farmed pacific white shrimp in three different areas of Ningde in 2022. The Kirby–Bauer disk method was used to detect *V. parahaemolyticus* resistance to 22 antibiotics, and resistant genes (such as quinolones (*qnr*VC136, *qnr*VC457, *qnr*A), tetracyclines (*tet* A, *tet*M, *tet*B), sulfonamides (*sul*I, *sul*II, *sul*III), aminoglycosides (*str*A, *str*B), phenicols (*cat*, *optr*A, *flo*R, cfr), β-lactams (*car*B), and macrolides (*erm*)) were detected by using PCR. The findings in this study revealed that *V. parahaemolyticus* was most resistant to sulfamoxazole, rifampicin, and erythromycin, with resistance rates of 56.9%, 36.3%, and 33.3%, respectively. Flufenicol, chloramphenicol, and ofloxacin susceptibility rates were 97.1%, 94.1%, and 92.2%, respectively. In all, 46% of the bacteria tested positive for multi-drug resistance. The virulence gene test revealed that all bacteria lacked the *tdh* and *trh* genes. Furthermore, 91.84% and 52.04% of the isolates were largely mediated by *cat* and *sul*II, respectively, with less than 5% resistance to aminoglycosides and macrolides. There was a clear mismatch between the antimicrobial resistance phenotypes and genotypes, indicating the complexities of *V. parahaemolyticus* resistance.

## 1. Introduction

Pacific white shrimp (*Litopenaeus vannamei*) belongs to the Decapoda order, Penaeidae family, and Penaediae Genus and is one of the most farmed shrimp species in the world [1]. Pacific white shrimp was introduced to China in 1988 and, since then, has been encouraged throughout the country in order to attain freshwater domestication and culture [2]. In the East China Sea region, Fujian Province is the primary farming area for pacific white shrimp. According to the most recent Fujian Province data, pacific white shrimp would account for around 94% of shrimp culture output in 2022, with a culture output of 21,636 tons and a culture area of 1259 hectares, delivering major economic benefits to Fujian Province [1]. However, as the size of aquaculture has expanded and the environment has worsened, the problem of pacific white shrimp aquaculture diseases has become more obvious, with bacterial infections caused by *Vibrio parahemolyticus* being one of the major issues restricting the industry’s growth.

*V. parahaemolyticus* is widely distributed in bays, offshore waterways, mudflats, and shellfish, and it is one of the most important diseases in pacific white shrimp aquaculture [3]. Recent research has revealed that *V. parahaemolyticus* with a particular plasmid(s) is the primary cause of acute hepatopancreatic necrosis disease (AHPND) outbreaks in shrimps [4]. AHPND can cause mass mortality in farmed shrimp, which has a negative impact on shrimp output and threatens the shrimp aquaculture industry [5]. At the moment, antibiotics are still an essential means of treating *V. parahaemolyticus* disease in shrimps since they can have a good therapeutic effect in the short term and greatly reduce the aquaculture industry’s economic losses. However, because of the widespread and long-term use of antibiotics in disease prevention, the species and number of drug-resistant *V. parahaemolyticus* are continually rising, potentially exacerbating the multi-drug-resistant issue [6,7].

*V. parahaemolyticus* is currently one of the most common pathogens of acute gastroenteritis in many coastal countries and regions worldwide, causing wound infection and sepsis. Food poisoning caused by *V. parahaemolyticus* is the second most prevalent cause of microbiological food poisoning in China [8,9]. The Ministry of Agriculture and Rural Affairs of China released the “Notice on Strengthening the Supervision and Regulation of Aquaculture Inputs” in January 2021, emphasizing categorically that the anarchy of aquaculture inputs has significantly jeopardized the quality and safety of aquaculture products. Serious challenges to the quality and safety of aquatic products must be addressed, as they jeopardize the healthy expansion of aquaculture to the point of intolerability. As a result, it is vital to expand *V. parahaemolyticus* resistance detection and resistance mechanism research, to lead the scientific prevention and management of *V. parahaemolyticus* disease in shrimp farming, and to protect human health.

By testing its sensitivity to common antibiotics and its resistance gene, Han et al. demonstrated that *V. parahaemolyticus* originating from pacific white shrimp in Guangxi Province was the most sensitive to florfenicol and the most resistant to sulfadiazine dipyridamole and Penicillin G [10]. Zhang et al. tested 21 strains of *V. parahaemolyticus* isolated from pacific white shrimp farms in the Shanghai area for antibiotic resistance, and the results revealed varying degrees of sensitivity to 14 common antibiotics, with enrofloxacin having the highest sensitivity and ampicillin/Penicillin, G/Sulfamethoxazole, and Penicillin G/Sulfamethoxazole having dominant resistance profiles [11]. Li et al. studied the treatment resistance of *V. parahaemolyticus* isolated from pacific white shrimp farm ponds in Shandong Province. They discovered that gentamicin, neomycin sulfate, and ampicillin resistance was the most severe, with resistance rates as high as 98%, 90%, and 86%, respectively, and that sensitivity to florfenicol, chloramphenicol, ceftazidime, and other antibiotics, as well as overall resistance, was more severe [12]. The majority of isolates exhibit different resistance profiles according to location, environment, and host conditions, while multi-drug resistance is becoming more common [13]. When drug-resistant bacteria infiltrate the human food chain, clinical antibiotic therapy will fail, posing a serious hazard to human health [14]. 

In this study, we systematically collected *V. parahaemolyticus* isolated from pacific white shrimp farms in Ningde regions (Xiapu, Fuding, and Jiaocheng), Fujian Province, from May to October and performed sensitivity studies on common antibiotics using the Kirby–Bauer disk diffusion method. Moreover, we detected the carriage of *V. parahaemolyticus* drug-resistant genes using PCR to analyze the correlation between the drug-resistant phenotype and the drug-resistant genes, with the goal of providing theoretical *V. parahaemolyticus* and revealing its drug resistance mechanism in order to give theoretical support.

## 2. Materials and Methods

### 2.1. Sample Collection

Samples of healthy growing and disease-free pacific white shrimp and their culture water were randomly collected from six farms in three cities (Xiapu, Fuding, and Jiaocheng) in Ningde regions, Fujian Province, respectively (Figure 1). All the samples were collected from May to October 2022.

### 2.2. V. parahaemolyticus Isolation 

About 25 g of pacific white shrimp were homogenized and inoculated into 225 mL of 3% alkaline peptone water (Oxiod, Basingstoke, UK) at 37 °C for at least 20 h. For the water samples, 25 mL of aquaculture water was mixed with 225 mL of 3% alkaline peptone water and incubated for at least 20 h at 37 °C. Following that, all samples were streaked on TCBS agar (Luqiao Co., Beijing, China) and incubated at 37 °C overnight. Following culture, the single colony was examined under Gram staining (Guangdong Huankai Microbial Technology Co., Guangzhou, China) and microscope (Olymus CX41RF, Tokyo, Japan) to determine its shape and structure before being stored at −80 °C in 15% glycerol.

### 2.3. DNA Extraction and Polymerase Chain Reactions (PCRs) 

The genomic DNA was extracted from potential V. parahaemolyticus colonies on a TCBS plate according to the manufacturer’s instructions using the Bacterial Genomic DNA Extraction Kit (Tiangen Biochemistry Technology Co., Beijing, China).

All the PCR reactions (Taq PCR MasterMix 10X, Takara, Kusatsu, Japan) were conducted in a 20 μL system. The primers and conditions are listed in Appendix A. The PCR products were sent to Sangong Bioengineering (Shanghai, China) Co., Ltd. for sequencing.

### 2.4. Antimicrobial Drug Susceptibility Test

The V. parahaemolyticus was coated with the entire MH plates (BD, Sparks, Philadelphia, PA, USA) and then pasted with K-B paper (Hangzhou Microbiology Reagent Co., Hangzhou, China). All the plates were incubated at 37 °C overnight. In total, 22 antibiotics were used in this study and listed in Appendix A.

Susceptibility% = (number of susceptible strains/total number of tested strains) × 100; whereas resistance rate% = (number of insensitive strains/total number of strains tested) × 100. 

### 2.5. Data Analysis

The sequence obtained was submitted to the basic local alignment search tool using BLAST at NCBI (National Center for Biotechnology Information) to determine the percentage similarity with already-identified 16S rRNA sequences in the GenBank database. The experimental data were analyzed and plotted using Microsoft software. The formula for calculating the coincidence rate between drug resistance phenotype and drug resistance gene is coincidence rate (%) = carry drug resistance gene and has corresponding drug resistance number of phenotypic strains/total number of strains with corresponding drug-resistant phenotype. 

## 3. Results

### 3.1. Distribution of V. parahaemolyticus

In this study, 102 strains of *V. parahaemolyticus* were isolated and identified from pacific white shrimp and water samples taken in the three cities (Xiapu, Jiaocheng, Fuding) belonging to Ningde regions in Fujian Province. Samples taken from September had the most strains (51 isolates) detected, while August had the fewest (three strains). *V. parahaemolyticus* was not discovered in Jiaodang, Zhujiabi, or Dajing culture water samples (Table 1); hence, the number of strains identified from pacific white shrimp was substantially greater than those recovered from water samples.

### 3.2. Drug Susceptibility Test Results

The isolated *V. parahaemolyticus* was tested for sensitivity to 22 common antibiotics, and the degree of resistance to different antibiotics varied among the strains, with sulfisoxazole having the highest rate of resistance (56.9%), followed by rifampicin (36.3%), erythromycin (33.3%), and streptomycin (32.4%). Furthermore, the majority of the other antimicrobial resistance rates were sensitive, with florfenicol having the highest sensitivity rate (97.1%), chloramphenicol (94.1%), and ofloxacin (92.2%) (Table 2). In China, all antimicrobials permitted for use in aquaculture have low resistance rates. Except for furazolidone (21.6%), resistance rates to norfloxacin, furotoxin, and chloramphenicol were all low among the aquaculture medications prohibited for use.

The multi-drug resistance distribution of 102 *V. parahaemolyticus* strains to 22 antimicrobial is depicted in Figure 2. The findings revealed that 47 strains of *V. parahaemolyticus* demonstrated multi-drug resistance (i.e., resistance to three or more antimicrobial drugs) in 46% (47/102) of the cases, with *V. parahaemolyticus* resistant to three drugs accounting for 16.7% (17/102) of the cases and 10 strains of *V. parahaemolyticus* resistant to more than ten antimicrobial drugs accounting for 9.8% of the cases.

The resistance rate of *V. parahaemolyticus* collected and isolated in Fuding to sulfisoxazole was 42.5%, followed by 32.5% and 20.0% resistance rates to rifampicin and streptomycin, respectively. Furthermore, most other antimicrobials had a resistance incidence of less than 20%, with 13 antibiotics not resistant to these antibiotics, including tetracycline, doxycycline, fluclobenicol, chloramphenicol, enrofloxacin, and ciprofloxacin. Resistance rates to sulfisoxazole and rifampicin antibiotics in *V. parahaemolyticus* collected in Xiapu were 62.5% and 43.8%, respectively, followed by flumequine (37.5%), erythromycin (34.4%), streptomycin (18.8%), furazolidone (15.6%), and others (Figure 3). The majority of *V. parahaemolyticus* sampled and isolated in Jiaocheng were resistant to sulfisoxazole, with a resistance rate as high as 87.5%, followed by streptomycin (62.5%), erythromycin (58.4%), rifampin (54.2%), and furazolidone (50.0%) which were all higher than 50%, and the resistance rate to cotrimoxazole, mephedrone, and polymyxin was 0 (Figure 3).

Figure 4 shows that Xiapu had 18 strains, of which 2 were seven-resistant, 3 were eight-resistant, and the highest was thirteen-resistant; Jiaocheng had 21 strains, with the 5 twelve-resistant strains accounting for 23.8% of the resistant strains in Jiaocheng; and Fuding had only 8 strains, with the highest being six-resistant.

### 3.3. Drug Resistance Gene Test Results

The carrier status of *V. parahaemolyticus* resistance genes is shown in Table 3. The detection rate for the lactam resistance gene *bla*CARB was 18.37% (18/98), whereas the detection rates for the sulfonamide resistance genes *sul*I, *sul*II, and *sul*III were 31.63% (31/98), 52.04% (51/98), and 19.39% (19/98, respectively). Aminoglycoside resistance genes strA and *str*B were discovered at rates of 4.08% (4/98) and 3.06% (3/98), respectively; tetracycl Quinolone resistance genes *qnr*VC136, *qnr*VC457, and *qnr*A were detected at rates of 27.55% (27/98), 1.02% (1/98), and 4.08% (4/98, respectively). *Cat*, *oprtr*A, *flo*R, and *cfr* resistance genes were found in 91.84% (90/98), 26.53% (26/98), 6.12% (6/98), and 6.12% (6/98), respectively. Moreover, 4.08% (4/98) of the macrolide resistance gene *erm* were found. 

### 3.4. Correlation between Drug Resistance Phenotype and Drug Resistance Genotype

According to Table 4, for the strains isolated in this experiment, the compliance rate between sulfisoxazole and *sul*I was 31%, 60.34% with *sul*II, and 18.97% with *sul*III; 40% with cotrimoxazole and *sul*I, 100% with *sul*II, and 40% with *sul*III; and 25% with *tet*A. The compliance rate between tetracycline and the *tet*A was 80%, 6.60%, respectively, with the neomycin and erythromycin resistance phenotype and the *str*B and the erm. Except for the resistance genes mentioned above, there was no association between the other resistance genes and the antibiotic resistance phenotype.

### 3.5. Virulence Gene Test Results

All *V. parahaemolyticus* isolates in this study were tested for the *tdh* and *trh* genes using PCR amplification, and the results showed that neither the *tdh* nor *trh* genes were found in any of the isolates.

## 4. Discussion

The problem of bacterial antibiotic resistance has received widespread attention, and the World Health Organization (WHO) identified it as one of the most serious dangers to human health in the twenty-first century [15]. In January 2022, *The Lancet* published the “Global Burden of bacterial Resistance 2019: The Systematic analysis report”, which provides the most comprehensive statistics on antibiotic resistance in its history and notes that in 2019, 4.95 million people died as a result of antibiotic failure. Moreover, the United Nations General Assembly explicitly declared in September that antimicrobial resistance (AMR) is already a global threat [16]. AMR also endangers the health of humans, animals, plants, and the environment, as well as the sustainability of agri-food systems. Meanwhile, AMR not only causes large economic losses in various industries but may also be transmitted to humans via the food chain, posing a serious threat to human food safety and health as well as a big therapeutic concern [17]. *V. parahaemolyticus*, one of the most dangerous foodborne pathogens, should be investigated for medicine resistance and pathogenicity. 

For antibiotic sensitivity testing in this study, 22 antimicrobial agents were chosen, including commonly used clinical pharmaceuticals in human medicine, national standard fishing drugs, veterinary drugs, and particularly, restricted antibiotics. The sensitivity of 102 isolates to 22 antimicrobial drugs indicated that the isolates exhibited clear resistances to sulfisoxazole, rifampicin, erythromycin, and streptomycin; relatively high susceptibility to chloramphenicol, fosfenicol, gentamicin, furotoxin, and cotrimoxazole; and resistance to the other medications of less than 30%. Many researchers have undertaken many studies on the treatment resistance of *V. parahaemolyticus* of shrimp origin in diverse locations in recent years. Zhang et al. tested the resistance of *V. parahaemolyticus* isolated from the Shanghai pacific white shrimp source in 2017–2020 and discovered that the resistance rate of these isolates to sulfamethoxazole was 71.43% [11]; Han et al. isolated *V. parahaemolyticus* samples sourced from adult pacific white shrimp culture pond from the coast of Qinzhou, China. *V. parahaemolyticus* resistance to sulfadimethoxine and sulfadiazine was 96.7% and 43.3%, respectively, much greater than resistance rates to other antibiotics [10]. 

Sulfonamides have been regularly used in aquaculture for 70 years and are one of the most important anti-infective drugs in the national standard fisheries pharmacopeia [18]. Sulfonamides help to reduce *V. parahaemolyticus*-induced diseases to some extent, but they also cause *V. parahaemolyticus*’ resistance to sulfonamides to increase year after year [17]. According to some studies, *V. parahaemolyticus* has a high resistance rate to sulfonamides, and the current concentration and detection rate of sulfonamides in China’s environment is much higher than in other countries. Sulfonamides present in aquaculture water, according to Wang et al., can promote *V. parahaemolyticus* resistance [19,20]. This implies that sulfonamides are no longer beneficial in the prevention and treatment of *V. parahaemolyticus*. Rifampicin, erythromycin, and streptomycin have high resistance rates, and it is probable that these two non-national standard fishery drugs are still in use or have been contaminated with pharmaceuticals or resistance genes.

Epidemiological studies have found a link between *V. parahaemolyticus* pathogenicity and hemolytic capacity, and its major hemolytic toxins are thermolysin intolerant hemolytic toxin, thermotolerant direct hemolytic toxin (*tdh*), and relatively thermotolerant direct hemolytic toxin (*trh*), which are encoded by the *tlh*, *tdh*, and *trh* genes, respectively [21]. The species-specific *tlh* gene is present in both clinical and environmental isolates. As a consequence, *tlh* may be used as a *V. parahaemolyticus* sub-target for molecular detection, and this experiment was validated with the *tlh* gene [22]. The major virulence components of *V. parahaemolyticus* are *tdh* and *trh*, and research has shown that environmental isolates seldom carry *tdh* and *trh* [23]. In this study, none of the isolates had *tdh* or *trh*. The isolates were likely obtained from shrimp breeding ponds with healthy growth and minimal disease, resulting in a low positive rate of virulence factors. Environmental and aquatic isolates with low *tdh* and *trh* carriage rates are mostly non-pathogenic or weakly pathogenic, and the WHO’s 2011 Risk Assessment of *V. parahaemolyticus* in seafood shows an increasing rate of clinical isolates that do not carry the *tdh* or *trh* genes but may carry other virulence factors that cause enterotoxicity and can result in severe cases [21,24]. Regardless of the low detection rate of pathogenic *V. parahaemolyticus* in shrimp and their farming environment, it should not be ignored.

*V. parahaemolyticus* was detected in shrimp and water samples cultured in three cities (Jiaocheng, Fuding, and Xiapu), which were investigated in this experiment as the main aquaculture areas of Ningde, but the resistance rate of *V. parahaemolyticus* in the Fuding samples was significantly lower than in the Jiaocheng and Xiapu samples. Fuding pacific white shrimp culture is primarily for the low-density mode of soil pond cultivation; the water ecology is better than the other two locations, and the dose of the drugs added in the daily aquaculture process is relatively low, so the white shrimp grows quickly and is not easily infected, according to the analysis of the three culturing methods. Jiaocheng and Xiapu are two sites where the high-density cultivation of high ponds is found. The farm likely uses more drugs in this form of culture than in the low-density culture mode of soil ponds, and the culture process is susceptible to *V. parahaemolyticus* and other ailments. The causes of this variety must be investigated further to see whether they are related to geographic position, distance, or cultural practices between culture regions or the distance between large sea areas that contribute to varying water quality conditions.

In this study, 46% of the *V. parahaemolyticus* isolates tested positive for multi-resistance to at least three antibiotics, with the bulk of the multi-resistant bacteria originating from Xiapu and Jiaocheng, which is likely related to the culture mode. The reasons for multi-drug resistance are more convoluted, and they may be linked to factors other than antibiotic usage. Nadella et al. revealed more than 61.1% of *V. parahaemolyticus* isolates from sick shrimp from various farms in many locations [25]. The multi-drug resistance of *V. parahaemolyticus* isolates from Zhejiang seafood is particularly alarming. Chen et al. reported that over 82% of the isolates were multi-drug-resistant to at least six drugs [26]. As a result, farmers should choose pharmaceuticals for shrimp and other aquatic animals in the aquaculture process based on drug sensitivity testing results in order to accomplish scientific drug usage.

According to research on the relationship between bacterial resistance phenotypes and resistance genotypes, the *sul* gene creates a dihydrofolate synthase, which leads to the loss of bacterial sensitivity to sulfonamides and induces bacterial resistance to sulfonamides [18]. The conformity rates between sulfisoxazole resistance phenotypes and *sul*I, *sul*II, and *sul*III genes were 31%, 60.34%, and 18.97%, respectively, and there was no perfect correspondence between *sul* and sulfonamide resistance phenotypes, which could be attributed to differences in the physiological conditions of the strains and the degree of the effective expression of the resistance genes. The most prevalent tetracycline resistance gene is *tet*A, and its expressed membrane protein preferentially pumps tetracyclines from the bacterial cytosol to extracellular compartments, resulting in bacterial resistance [27]. Despite the presence of 42 *tet*A genes in 98 strains, only 14.7% and 11.8% of the tested strains were resistant to tetracycline and doxycycline, respectively. In addition to the *tet*A gene-mediated efflux pump resistance mechanism, other methods of tetracycline resistance exist, such as *tet*M and *tet*B gene-mediated tetracycline blunting and ribosome protection [28]. Only 2 of the 19 *V. parahaemolyticus* in the current study that had the *tet*M gene were mediated by doxycycline in addition to only 1 by tetracycline, demonstrating that there is not a complete match between resistance genes and tetracycline-resistant phenotypes. As a consequence, other mechanisms of *V. parahaemolyticus* resistance to tetracycline medicines in Ningde white shrimp need further investigation.

In this study, 102 strains of *V. parahaemolyticus* were isolated from shrimp culture ponds in Ningde regions. The results of the drug sensitivity test showed that *V. parahaemolyticus* had the most serious drug resistance to sulfamisoxazole and rifampicin, with resistance rates of 56.9%, 36.3%, and 33.3%, respectively. It was highly sensitive to flufenicol, chloramphenicol, and ofloxacin, with sensitivity rates as high as 97.1%, 94.1%, and 92.2%. 46.0% of the strains were multi-drug resistant. Virulence gene test results showed that all strains did not carry the *tdh* and *trh* genes. The detection rate of ***cat*** was up to 91.84%. The transverse amine resistance gene *sul*II was the second, and the detection rate was 52.04%. The detection rates of aminoglycoside and macrolide resistance genes were very low, both less than 5%. There was no one-to-one correlation between the detection rate of drug resistance genes and the drug resistance phenotype, indicating the complexity of the drug resistance of *V. parahaemolyticus*, providing a theoretical basis for the scientific prevention and control of *V. parahaemolyticus* of pacific white shrimp in Ningde regions and revealing its drug resistance mechanism.

## Figures and Tables

**Figure 1 microorganisms-12-00152-f001:**
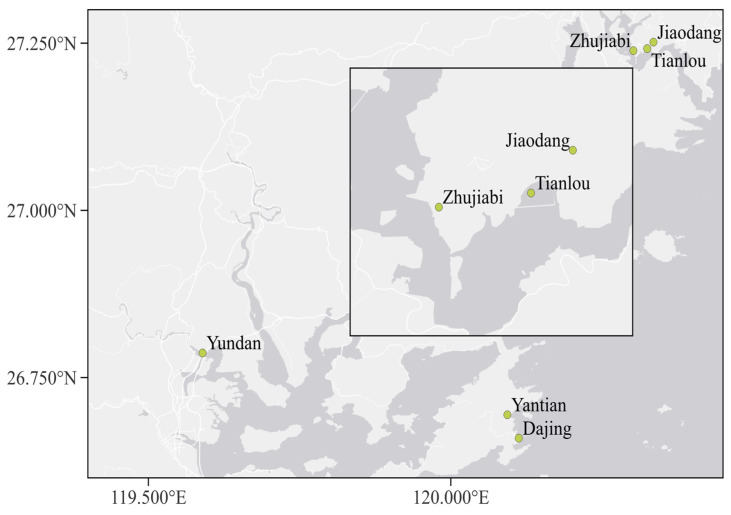
All samples collected from six farms in three cities belonging to Ningde regions. Farm Yundan is from Jiaochen; two farms, Yantian and Dajiang, are from Xiapu; and three farms, Jiaodang, Zhujiabi, and Tianlou, are from Fuding.

**Figure 2 microorganisms-12-00152-f002:**
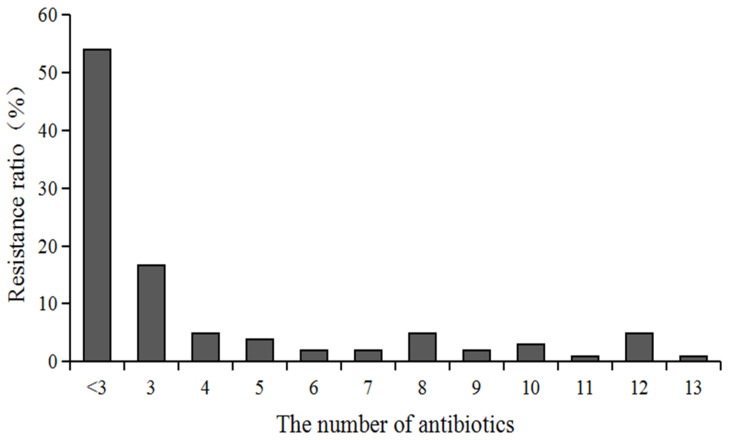
Multi-drug resistance distribution of all *V. parahaemolyticus* isolates to 22 antibiotics.

**Figure 3 microorganisms-12-00152-f003:**
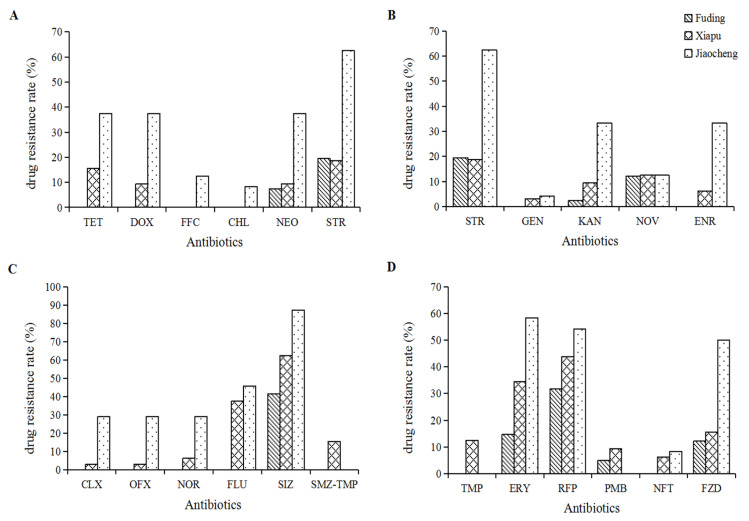
Resistance rates of all *V. parahaemolyticus* isolated from three cities in Ningde to 22 antibiotics. Rssistance rates of all *V. parahaemolyticus* isolates to Tetracycline (TET), doxycycline (DOX), flufenicol (FFC), chloramphenicol (CHL), neomycin (NEO), streptomycin (STR) present at subfigure (**A**), to gentamicin (GEN), kanamycin (KAN), norfloxacin (NOV) and enrofloxacin (ENR) prensent at subfigure (**B**), to ciprofloxacin (CLX), oxofloxacin (OFX), nofloxacin (NOR), flimequine (FLU), sulfamisoxazole (SIZ), timethoprim/sulfamethoxazole (SMZ-TMP) present at subfiure (**C**), and to trimethoprim (TMP), erythromycin (ERY), rifampin (RFP), polymyxin B (PMB), nitrofurantoin (NFT), furazolidone (FZD) present at subfigure (**D**), respectively.

**Figure 4 microorganisms-12-00152-f004:**
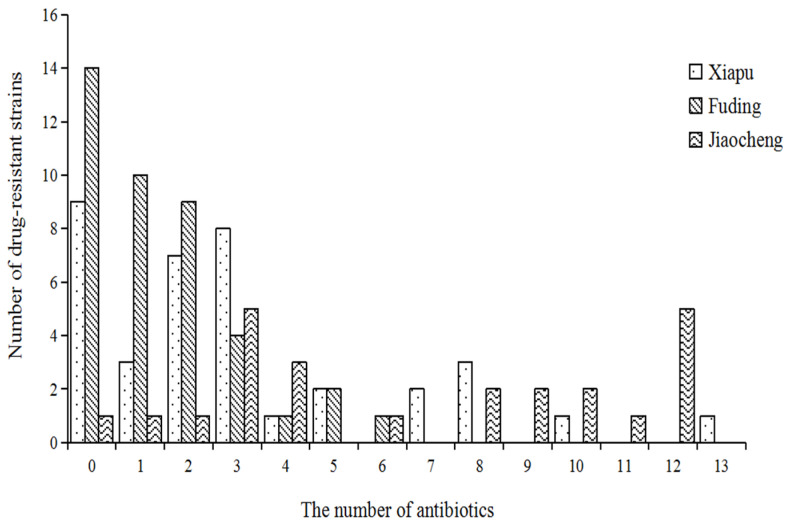
Number of multi-drug-resistant strains of *V. parahaemolyticus* isolated from Xiapu, Fuding, and Jiaocheng.

**Table 1 microorganisms-12-00152-t001:** *V. parahaemolyticus* found in three cities in Ningde, collected from May to October.

Location		Samples Collected	Months						Total
			May	June	July	August	September	October	
Fuding	Jiaodang	shrimp	3	0	0	0	7	0	10
		water	0	0	0	0	0	0	0
	Tianlou	shrimp	0	0	0	0	10	0	10
		water	2	0	0	0	0	0	2
	Zhujiabi	shrimp	7	0	0	0	12	0	19
		water	0	0	0	0	0	0	0
Jiaocheng	Yundan	shrimp	0	4	5	2	4	1	16
		water	0	1	6	1	0	0	8
Xiapu	Yantian	shrimp	2	10	0	0	8	3	23
		water	0	2	0	0	9	0	11
	Dajing	shrimp	0	0	0	0	1	2	3
		water	0	0	0	0	0	0	0
Total			14	17	11	3	51	6	102

**Table 2 microorganisms-12-00152-t002:** Numbers of *V. parahaemolyticus* resistant to 22 antibiotics and their resistance rates.

Antibiotics	Numbers of Resistance Isolates	Antibiotics Resistance Rate (%)	Sensitivity Rate (%)
Tetracycline (TET)	15	14.7	62.7
Doxycycline (DOX)	12	11.8	77.5
Flufenicol (FFC)	3	2.9	97.1
Chloramphenicol (CHL)	2	2.0	94.1
Neomycin (NEO)	15	14.7	29.4
Streptomycin (STR)	33	32.4	38.2
Gentamicin (GEN)	3	2.9	74.5
Kanamycin (KAN)	13	12.7	42.2
Norfloxacin (NOV)	12	11.8	42.2
Enrofloxacin (ENR)	10	9.8	50.0
Ciprofloxacin (CLX)	8	7.8	76.5
Oxofloxacin (OFX)	8	7.8	92.2
Norfloxacin (NOR)	9	8.8	83.3
Flumequine (FLU)	16	15.7	53.9
Sulfamisoxazole (SIZ)	58	56.9	22.5
Trimethoprim/sulfamethoxazole (SMZ-TMP)	5	4.9	86.3
Trimethoprim (TMP)	9	8.8	80.4
Erythromycin (ERY)	34	33.3	3.9
Rifampin (RFP)	37	36.3	21.6
Polymyxin B (PMB)	9	8.8	47.1
Nitrofurantoin (NFT)	4	3.9	76.5
Furazolidone (FZD)	22	21.6	62.7

**Table 3 microorganisms-12-00152-t003:** The carrier status of *V. parahaemolyticus* resistance genes.

Category	Antibiotics Resistance Genes	Number of Isolates Carry Resistant Genes	Resistance Gene Carrier Rate (%)
Quinolones	*qnr*VC136	27	27.55
	*qnr*VC457	1	1.02
	*qnr*A	4	4.08
Tetracyclines	*tet*A	42	42.86
	*tet*M	19	19.39
	*tet*B	12	12.24
Sulfonamides	*sul*I	31	31.63
	*sul*II	51	52.04
	*sul*III	19	19.39
Aminoglycosides	*str*A	4	4.08
	*str*B	3	3.06
Macrolide	*erm*	4	4.08
Amphenicols	*cat*	90	91.84
	*optr*A	26	26.53
	*flo*R	6	6.12
	*cfr*	6	6.12
Beta-lactams	*bla*CARB	18	18.37

**Table 4 microorganisms-12-00152-t004:** Correlation between antibiotic resistance and resistance genes of *V. parahaemolyticus* of white shrimp in Ningde regions.

Antibiotics	Antibiotic-Resistance Genes	Compatibility between Resistance Phenotype and Resistance Genes (%)
SIZ	*sul*I	31.00
	*sul*II	60.34
	*sul*III	18.97
SMZ-TMP	*sul*I	40.00
	*sul*II	100.00
	*sul*III	40.00
DOX	*tet*A	83.33
	*tet*M	25.00
TET	*tet*A	80.00
	*tet*B	6.60
	*tet*M	20.00
NEO	*str*B	6.67
ERY	*erm*	2.94

## Data Availability

Data are contained within the article and Appendix A.

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
