# Peer review of "Antibiotic Resistance and Genetic Profiles of Vibrio parahaemolyticus Isolated from Farmed Pacific White Shrimp (Litopenaeus vannamei) in Ningde Regions"

_microorganisms, 2024, doi:10.3390/microorganisms12010152_

Round 1

Reviewer 1 Report

Comments and Suggestions for Authors

The manuscript is very well written and well-organized and with results that could be vital for the decrease of antibiotic resistance, also having important economic importance.

I would have a small comment regarding the food for cultured shrimp, if its composition is known and if it contains antibiotics to prevent the white shrimp from getting sick.

Please complete this aspect.

Author Response

Dear Editor,

On behalf of all writers, I would like to convey my appreciation for the reviewers' remarks on the manuscript. Their remarks greatly enhance the manuscript. I would also want to answer their questions as follows.

  • I would have a small comment regarding the food for cultured shrimp, if its composition is known and if it contains antibiotics to prevent the white shrimp from getting sick.

A: Thanks for your comment. Antibiotics are the most extensively used antimicrobial agents because they are inexpensive and effective. As a result, antibiotics are administered to keep the white shrimp from getting sick. Since the long-term usage of antibiotics, resistance has been discovered regularly. Scientists have started producing disease-resistant white shrimp using genomic breeding techniques. Hopefully, no antibiotic-resistant white shrimp will be on our customers' tables soon.

Reviewer 2 Report

Comments and Suggestions for Authors

This manuscript is a comprehensive study of bacterial resistentance exhibited by Vibrio parahaemolyticus. The reserahcers isolated this specie in three differnt regions and tested their resistance to a broad range of antibiotics. This research has great practical and fundamental importance.

SOme minor comments should be addressed:

1) Why you have not evaluated the resustance towards other bactericidal agents, such as silver ions, Zn, etc? What would be the effect?

2) Please explicitly draw the limitations of your currect methodological approach.

Author Response

Dear Editor,

On behalf of all writers, I would like to convey my appreciation for the reviewers' remarks on the manuscript. Their remarks greatly enhance the manuscript. I would also want to answer their questions as follows.

  • Why you have not evaluated the resustance towards other bactericidal agents, such as silver ions, Zn, etc? What would be the effect?

A: Thanks for your comment. Antibiotics are the most widely used antimicrobial agents due to their low cost and efficacy. Of course, metals like Au, Ag, and others have antibacterial properties, but they also have drawbacks; for example, a certain amount of Ag is harmful to humans. At the same time, gold, silver, and other metals are significantly more expensive. They are hence impractical for usage.

  • Please explicitly draw the limitations of your currect methodological approach.

A: Thanks for your comment. The current approach to the investigation is based on PCR and medium plating for Vibiro parahaemolyticus detection. However, some limits still exist. For example, 1-Plating detection was determined by colony isolation, chemical analysis, and so on; it may take 48 to 72 hours to finish the entire method, which is time demanding; 2-PCR-based detection occasionally produces false-positive findings. Advanced technology is necessary.